# Comprehensive Characterization of Lignans from *Forsythia viridissima* by UHPLC-ESI-QTOF-MS, and Their NO Inhibitory Effects on RAW 264.7 Cells

**DOI:** 10.3390/molecules24142649

**Published:** 2019-07-22

**Authors:** Jungmoo Huh, Chang-Min Lee, Seoyoung Lee, Soeun Kim, Namki Cho, Young-Chang Cho

**Affiliations:** 1College of Pharmacy, Seoul National University, Gwanak-gu, Seoul 08826, Korea; 2Department of Laboratory Medicine, College of Veterinary Medicine, Chonnam National University, Gwangju 61186, Korea; 3College of Pharmacy, Chonnam National University, Gwangju 61186, Korea

**Keywords:** *Forsythia viridissima*, lignan dimers, lignans, lignan glycosides, LC-MS^e^ analysis, anti-inflammatory activity, RAW 264.7

## Abstract

Lignans are known to be an important class of phenylpropanoid secondary metabolites. In the course of our studies on the chemodiversity of lignans, the necessity arose to develop a method for the fast detection and identification of bioactive lignan subclasses. In this study, we detected 10 lignan derivatives of different extracts of *F. viridissima* by UHPLC-ESI-QTOF-MS. Lignan glycosides (**1** and **2**), lignans (**3** and **4**), and lignan dimers (**5**–**10**) were identified by analysis of their exact masses and MS^e^ spectra along with the characteristic mass fragmentation patterns and molecular formulas. We further investigated NO inhibitory effects of *F. viridissima* fractions and their major lignan derivatives to evaluate those anti-inflammatory effects. The methylene chloride fraction of *F. viridissima* as well as compounds **8** and **10** showed potent dose-dependent NO inhibitory effects on RAW 264.7 cells. Corresponding to the NO inhibition by compounds **8** and **10**, lipopolysaccharide (LPS)-induced inducible nitric oxide synthase (iNOS) expression was notably reduced by both compounds. Our combined data with the bioactive results and the component analysis by UHPLC-ESI-QTOF-MS suggest that the methylene chloride fraction of *F. viridissima* roots could be potential anti-inflammatory agents and these are related to major lignans including dimeric dibenzylbutyrolactone lignans.

## 1. Introduction

Lipopolysaccharide (LPS), the major cell wall component of gram-negative bacteria, induces inflammatory responses when administered to cells or animals. It induces the production of inflammatory mediators, including nitric oxide (NO), prostaglandin E_2_, and proinflammatory cytokines [1,2,3,4]. Although enhanced production of inflammatory mediators is important for host defense against external stimuli including LPS, excess production of inflammatory mediators causes severe inflammatory diseases, including septic shock, rheumatoid arthritis, systemic lupus erythematosus (SLE), and inflammatory bowel disease (IBD) [5,6,7]. Therefore, an agent that alleviates excess amounts of inflammatory mediators could be applied to treat various inflammatory diseases. Although various anti-inflammatory drugs, such as non-steroidal anti-inflammatory drugs (NSAIDs), have been developed, various natural products and those components are under evaluation for the development of new anti-inflammatory drugs due to the severe adverse effects of NSAIDs [8].

‘Yeon-kyo’ is the fruit of *Forsythia viridissima* (Oleaceae) and *F. suspensa*, listed in the 11^th^ edition of the Korean Pharmacopoeia (KP11). Arctigenin is known as an indicator component for the quantification of *F. viridissima*. Lignan is one of the representative secondary metabolite class in *F. viridissima*. Other lignans, such as matairesinol, arctiin, and matairesinoside [9], have been reported as constituents of *F. viridissima* along with phenylethanoid glycosides, flavonoids, and triterpenoids [10,11]. Traditionally, ‘Yeon-kyo’ has been used for antiviral, anti-inflammatory, diuretic, antimicrobial, detoxification, and antipyretic activities. Among the different biological activities, the anti-inflammatory activity as a representative effect has been shown by numerous in vivo and in vitro studies with *F. suspensa* and *F. viridissima* [12,13]. Arctigenin, a phytochemical marker of *F. viridissima* [14], and other lignans [15] are also known to exert anti-inflammatory activities. Previously, we reported six new dimeric lignans and one new lignan glycoside along with nine known lignans from the roots of *Forsythia viridissima* [16]. Based on our previous research, this study aimed to investigate the profiles of the lignan dimers, lignans, and lignan glycosides with the help of a mass spectrometric technique and to assess the anti-inflammatory activities of isolated compounds of roots of *F. viridissima*.

## 2. Results and Discussion

Dried roots of *F. viridissima* (2.7 kg) were extracted with 80% aqueous MeOH (3 times × 4 L, 90 min, 25 °C) by ultrasonication, and the crude extracts were diluted in H_2_O and partitioned successively with *n*-hexane, CH_2_Cl_2_, and *n*-BuOH. Compounds **1**–**10** for in vitro assay were isolated from CH_2_Cl_2_ fraction of the roots of *F. viridissima* by using chromatographic methods including HPLC, MPLC over C_18_ RP column as previously described [16]. The specific lignans of CH_2_Cl_2_ and *n*-BuOH fractions were identified by UHPLC-ESI-QTOF-MS^e^ analysis in both positive and negative ion modes. Lignan glycosides (**1** and **2**), lignans (**3** and **4**), and lignan dimers (**5**–**10**) were identified in total, CH_2_Cl_2_, and *n*-BuOH fractions in both positive and negative ion modes. A total of 10 compounds was detected, and their structures are shown in Figure 1, and their retention time (Rt), error (ppm), molecular ions, and molecular formulas are shown in Table 1.

### HR-MS Characterization of Lignan and Lignan Glycosides

The lignans present in *F. viridissima* were identified by calculation of the molecular formula from exact mass and MS^e^ spectra with the characteristic mass fragmentation patterns. The lignan glycosides, matairesinoside (**1**) and arctiin (**2**) were detected mainly in the *n*-BuOH fractions of *F. viridissima* (Appendix A). Matairesinoside and arctiin (*m*/*z,* 543.1844 and 557.2019 [M + Na]^+^) (Table 1) and their derivative (*m*/*z,* 717.2894 and 745.3224 [2M + H]^+^), and loss of the glucose portion (*m*/*z,* 359.1494 and 373.1662 [M + H − 162]^−^) were found along with typical fragment ions *m*/*z* 137 and/or 151 of the dibenzylbutyrolactone lignan (Appendix A). 

The MS^e^ spectra of dimatairesinol (**5**) showed molecular ions *m*/*z* 715.2741 ([M + H]^+^), 737.2561 ([M + Na]^+^), 697.2642 ([M + H − H_2_O]^+^), and 679.2540 ([M + H − 2H_2_O]^+^, which are found in the dichloromethane fraction of *F. viridissima* (Appendix A). The characteristic MS fragment of *m*/*z* 137 of matairesinol (Appendix A) was also observed in agreement with [17,18].

Both viridissimaol A (**6**) and B (**7**) detected in the dichloromethane fraction were lignan dimers with matairesinol and arctigenin moieties, and therefore, showed similar fragment ion peaks. Viridissimaol A (**6**), the protonated ion peak and the addition of the sodium ion were detected at *m*/*z* 729.2905 and 751.2716, respectively, and loss of one or two water molecules corresponded at *m*/*z* 711.2792 or 693.2686 (Figure 2 and Appendix A). Similarly, the fragment ion peaks of viridissimaol B (**7**) were detected at *m*/*z* 729.2896 and 751.2712, which exhibited [M + H]^+^ and [M + Na]^+^ (Appendix A). The daughter ions of both the compounds also exhibited diagnostic ions at *m*/*z* 137 and 151. 

Viridissimaol E (**8**) is a dimer associated with arctigenin and (7*E*,8′*R*)-7,8-didehydroarctigenin, which was detected in the dichloromethane fraction (Appendix A). Viridissimaol E (*m*/*z* 741.2907 [M + H]^+^) showed its derivative 763.2717 ([M + Na]^+^) and loss of water molecules 723.2800 ([M + H − H_2_O]^+^) and 705.2671 ([M + H − 2H_2_O]^+^) (Appendix A). Since compound **8** contained arctigenin, it showed a common MS fragment value of *m*/*z* 151. In contrast, the observed fragmentation at *m*/*z* 247 could occur due to the unsaturation of the position 7 and 8 followed by a rearrangement resulting in loss of the substituted benzene moiety (Figure 3) [17,18]. 

Diarctigenin (**9**, *m*/*z* 743.3055 [M + H]^+^) and conicaol A (**10**, *m*/*z* 743.3060 [M + H]^+^), which belong to the dichloromethane fraction (Appendix A) have partial structures of arctigenin, and both compounds showed an MS fragment *m*/*z* 151. Diarctigenin (**9**) showed molecular ions *m*/*z* 743.3055 ([M + H]^+^), 765.2867 ([M + Na]^+^), 725.2947 ([M + H − H_2_O]^+^), and 707.2844 ([M + H − 2H_2_O]^+^ (Appendix A). In addition, the MS^e^ spectra of conicaol A were obtained as *m*/*z* 743.3060 ([M + H]^+^), 765.2891 ([M + Na]^+^), 725.2958 ([M + H − H_2_O]^+^), and 711.2811 ([M + H − 2H_2_O]^+^ (Appendix A).

To elucidate whether *F. viridissima* roots exert anti-inflammatory effects through the inhibition of proinflammatory responses, we performed a cell viability assay and an NO assay using single components from *F. viridissima* roots in LPS-unstimulated and LPS-stimulated RAW 264.7 macrophage, a murine macrophage cell line which secretes inflammatory mediators through activation with TLR ligands such as LPS. Since the anti-inflammatory effects of the compounds are important when these effects are significant at cytotoxic concentration, we attempted to validate the non-cytotoxic concentrations of the compounds in RAW 264.7 macrophages. As shown in Figure 4, only compounds **3** (at 50 and 100 µM) and **9** (at 100 µM) showed significant cytotoxicity, whereas the other compounds did not show cytotoxicity at the highest dose of each compound. Based on the cell viability, the alleviation of NO production by the compounds in LPS-stimulated macrophages was estimated at non-cytotoxic concentrations. Of all compounds, compounds **3**, **6**, **7**, **8**, and **10** notably inhibited LPS-mediated production of NO in a dose-dependent manner. Among them, dibenzylbutyrolactone lignans, we selected two compounds which have slightly showed more inhibitory activity and have a different part of molecules. Compounds **8**, which have 7, 8 unsaturated double-bond, and compound **10**, which have C-O-C bond linkage in dimer structure, represent anti-inflammatory potential among lignan subclass from the methylenechloride fraction of *F. viridissima* roots. To verify that the reduced production of NO by compounds **8** and **10** was due to the transcriptional regulation of iNOS, an enzyme responsible for the production of NO, the iNOS protein expression level was measured by immunoblotting. Corresponding to the NO inhibition by compounds **8** and **10**, the LPS-induced iNOS expression was notably reduced by these compounds (Figure 5). These data prompted us to estimate the effects of the methylene chloride fraction on NO production in LPS-stimulated RAW 264.7 macrophages since compounds **8** and **10** were prepared in the methylene chloride fraction. As expected, the methylene chloride fraction notably inhibited the LPS-stimulated NO production without showing any cytotoxicity (Figure 6).

To date, to the best of our knowledge, there are few reports aimed at assessing the in vitro biological effects or identifying the complex bioactive lignans from the root parts of *F. viridissima*. In our present study, we narrow it down to study chemical analysis including dibenzylbutyrolactone dimer lignans by using UHPLC-ESI-QTOF-MS method, since the methylene chloride fraction of *F. viridissima* roots showed potent NO inhibitory effects in our experimental systems. Taken together, combined with the bioactive results and the component analysis by UHPLC-ESI-QTOF-MS, we suggest that the roots of *F. viridissima*-mediated anti-inflammatory effects are related to major lignans including dimeric dibenzylbutyrolactone lignans. 

## 3. Materials and Methods 

### 3.1. Plant Materials

*Forsythia viridissima* roots were collected in June 2015 from the Medical Herb Garden, College of Pharmacy, Seoul National University, Goyang-si, Gyeonggi-do, Korea. *F. viridissima* was identified by S. I. Han (Medical Herb Garden, College of Pharmacy, Seoul National University).

### 3.2. Chemicals and Reagents

All tested compounds were isolated from CH_2_Cl_2_ fraction of *F. viridissima* as described previously [16]. All isolates were lyophilized to remove any solvent that may be present. After dissolution in dimethyl sulfoxide (DMSO) for use in cell culture, it was diluted to a suitable concentration. LPS and DMSO were purchased from Sigma-Aldrich. LPS was dissolved in phosphate-buffered saline (PBS). Mouse anti-glyceraldehyde 3-phosphate dehydrogenase (GAPDH) was purchased from Santa Cruz Biotechnology (Santa Cruz, CA, USA). Mouse anti-iNOS was purchased from BD Biosciences (Franklin Lakes, NJ, USA).

### 3.3. Chromatographic Profiling of the Compounds Present in F. viridissima Subfractions

The analysis of the dichloromethane subfractions was performed on a Waters Xevo G2 Q-TOF mass spectrometer (Waters MS Technologies, Manchester, UK), which was equipped with an electrospray ionization interface with Waters ACQUITY UHPLC system (Waters, Co., Milford, MA, USA). The UHPLC-MS data were obtained by MassLynx 4.1 software (Waters, UK). The separation of the compounds was carried out on ACQUITY UHPLC^®^ BEH C18 column (100 × 2.1 mm, 1.7 μm, Waters Co.). The mobile phase was composed of 0.1% (*w*/*v*) formic acid in water (solvent A) and acetonitrile (solvent B) in the following gradients; 0 min (A:B 90:10), 8 min (A:B 40: 60), 12 min (A:B 0:100), 15 min (A:B 0:100), 15.1 min (A:B 90:10), and 18 min (A:B 90:10). Injections were carried out with an autosampler (4 °C), the injection volume was 1 μL, and the column temperature was 40 °C. The flow rate was 0.3 mL/min for a total run time of 18 min. The ESI parameters were as follows: negative ion mode, positive ion mode; cone gas flow rate, off; cone voltage, 40 V; source temperature, 120 °C; desolvation gas flow, 600.0 L/h; and capillary voltage, 3.0 kV. The collision gas was argon, and the nebulizer and auxiliary gas was high purity nitrogen gas. Leucine enkephalin (*m/z* 554.2615, [M − H]^−^, 556.2771 [M + H]^+^) was used as the lock mass. The collision energy was set to 20 to 40 eV.

### 3.4. Cell Culture

RAW 264.7 macrophage, a mouse monocyte cell line, was obtained from ATCC (Manassas, VA, USA). Cells were incubated in Dulbecco’s Modified Eagle’s Medium (DMEM; Invitrogen, Carlsbad, CA, USA), supplemented with 10% fetal bovine serum (FBS; Invitrogen) and antibiotics (100 U/mL penicillin and 100 µg/mL streptomycin; GIBCO BRL, Grand Island, NY, USA), at 37 °C in a humidified air incubator with 5% CO_2_. 

### 3.5. Cell Viability Assay

RAW 264.7 cells (6.0 × 10^4^ cells/well) were seeded into a 96-well plate. After overnight incubation, cell culture medium was removed, various concentrations of compound diluents were applied to the cells, and then incubated for 24 h. Same volume of DMSO was treated to compound-untreated group to exclude the effect of DMSO on cell viability. After incubation, the cells were treated with EZ-Cytox solution (DAEIL lab, Seoul, Korea; 1/10 of the culture medium) for additional 1 h at 37 °C. Cell viability was determined by measuring the absorbance of the resulting formazan product at 450 nm using a VersaMax Microplate Reader (Molecular Devices, LLC, Silicon Valley, CA, USA).

### 3.6. Nitrite Assay

Nitrite assay is based on the fact that measurement of nitrite (NO_2_^−^), which is one of two primary, stable and nonvolatile breakdown products of NO, reflects the quantity of NO in the supernatant [19]. RAW 264.7 cells (6.0 × 10^4^ cells/well) were seeded into a 96-well plate and incubated overnight for cell adhesion. After adhesion, compounds were applied to the cells with indicated concentrations, LPS (1 µg/mL) was subsequently treated to the cells, and cells were then incubated for 24 h. After 100 µL cultured media was transferred to a new 96-well plate, 100 µL Griess reagent (a mixture of 1% sulfanilamide, 2.5% phosphoric acid (H_3_PO_4_), and 0.1% N-(1-naphthyl) ethylenediamine in distilled water) was added to each well [19]. Sodium nitrite was serially diluted from 64 µM to 1 µM and a standard curve was generated by measuring absorbance after application of Griess reagent. The absorbance at 540 nm was determined by a VersaMax Microplate Reader.

### 3.7. Immunoblotting

RAW 264.7 cells were treated with compounds **8** and **10** (5 and 25 µM) following LPS (1 µg/mL) stimulation for 24 h; subsequently, the cells were washed with ice-cold PBS. Cells were lysed in lysis buffer containing 0.5% NP-40, 0.5% Triton X-100, 150 mM NaCl, 20 mM Tris-HCl (pH 8.0), 1 mM ethylenediaminetetraacetic acid, 1% glycerol, 10 mM NaF, 1 Na_3_VO_4_, and 1 mM phenylmethylsulfonyl fluoride (PMSF), and were centrifuged at 15,814 × g for 30 min at 4 °C. Supernatants were used for immunoblotting analyses. Same amounts of cell lysates were separated on 10% sodium dodecyl sulfate-polyacrylamide gels (SDS-PAGE) and transferred to a nitrocellulose membrane (GE Healthcare, Milwaukee, WI, USA) using wet blotting method. The membrane was blocked for 1 h with 5% nonfat-dried milk in Tris-buffered saline with 0.05% Tween-20 (TBST) buffer, followed by overnight incubation at 4 °C with 1:1000 dilution of primary antibodies. Each membrane was further incubated for 2 h with secondary peroxidase-conjugated goat immunoglobulin G (IgG; 1:5000). The target proteins were detected using an enhanced chemiluminescence (ECL) solution. Protein levels were quantified by scanning the immunoblots and analyzing them using LabWorks software (UVP Inc., Upland, CA, USA).

### 3.8. Statistical Analysis and Experimental Replicates

Cell viability assay, nitrite assay, and immunoblotting were repeated three times. Each result was represented as mean ± standard deviation (SD). Differences between experimental conditions were assessed by Student’s *t*-test, performed by Prism 3.0 (Graph Pad Software, San Diego, CA, USA), and * *p* < 0.05 was considered statistically significant.

## Figures and Tables

**Figure 1 molecules-24-02649-f001:**
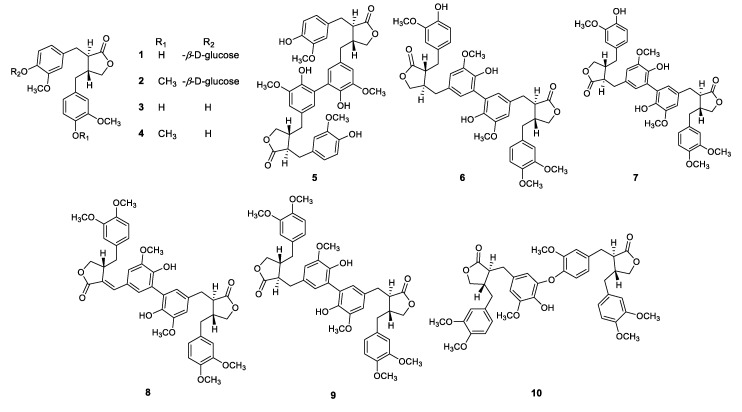
All isolated compounds (**1**–**10**) from *Forsythia viridissima*.

**Figure 2 molecules-24-02649-f002:**
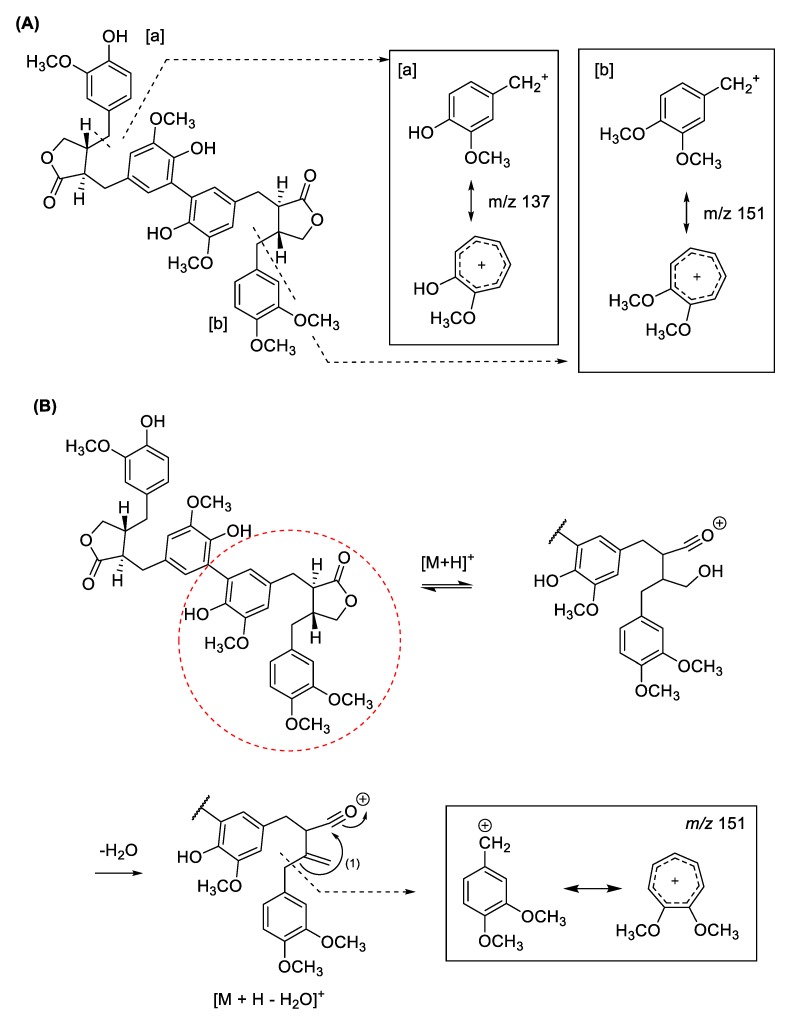
MS fragmentation pattern of protonated viridissimaol A (**6**). (**A**) Characteristic MS fragment value *m*/*z* 137 (**a**) and 151 (**b**) (**B**) MS fragmentation pathways of protonated viridissimaol A.

**Figure 3 molecules-24-02649-f003:**
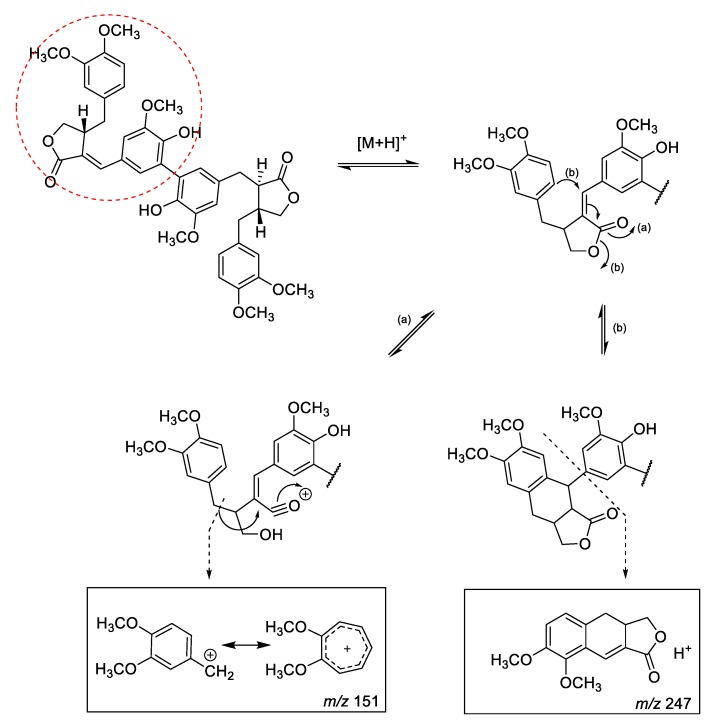
Characteristic ESI/MS fragmentation (highlighted part) of viridissimaol E (**8**).

**Figure 4 molecules-24-02649-f004:**
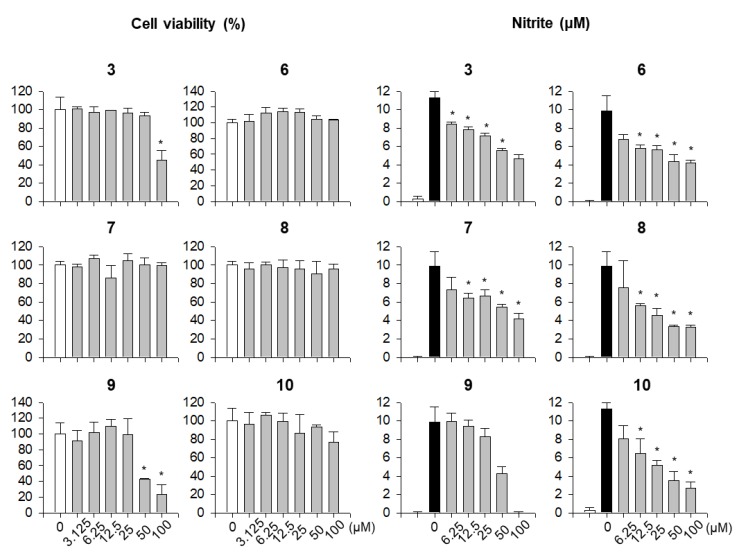
Cytotoxic and NO inhibitory effects of compounds (**3**, **6**, **7**, **8**, **9**, and **10**). The cell viability data were expressed as relative values to the untreated control group. NO levels were calculated according to a standard curve plotted using nitrite standard solution. Data represent the mean ± SD. * *p* < 0.05 relative to the control group (untreated group for cell viability and lipopolysaccharide (LPS)-treated group for nitrite level). Each number above graph is the compound number used for experiments. White, black, and gray bar represent Untreated, LPS-treated, and compound-treated groups, respectively. Experiments performed three times independently in three replicates for each experiment.

**Figure 5 molecules-24-02649-f005:**
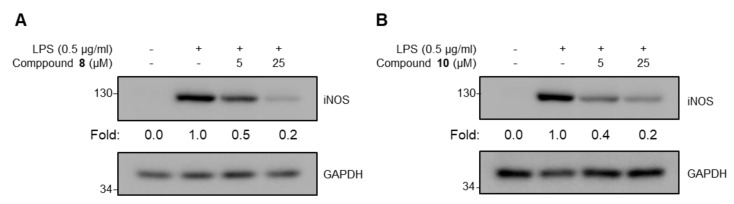
Inhibitory effect of compounds **8** (**A**) and **10** (**B**) on the inducible nitric oxide synthase (iNOS) protein expression. Experiments performed three times independently (*n* = 3). A representative data and fold induction (relative expression level by comparing with LPS-treated group as 1.0) was shown.

**Figure 6 molecules-24-02649-f006:**
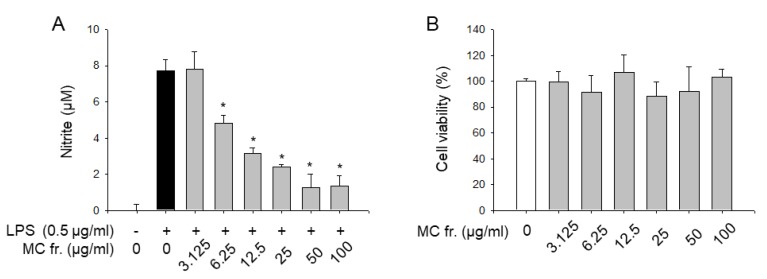
NO inhibitory effect of CH_2_Cl_2_ fraction of *Forsythia viridissima* roots. (**A**) NO levels were calculated according to a standard curve plotted using nitrite standard solution. (**B**) The cell viability data were expressed as relative values to the untreated control group. Data represent the mean ± SD. * *p* < 0.05 relative to the control group (untreated group for cell viability and LPS-treated group for nitrite level). White, black, and gray bar represent Untreated, LPS-treated, and compound-treated groups, respectively. Experiments performed three times independently in three replicates for each experiment.

**Table 1 molecules-24-02649-t001:** Compounds identified by molecular formulas calculated from the accurate mass and MS^e^ fragments of UHPLC-ESI-QTOF-MS analyses.

Rt (min)	[M + H]^+^ ([M + Na]^+^)	Molecular Formula	Error (ppm)	MS Fragments (*m*/*z*)	Compound
4.03	(543.1844)	C_26_H_43_O_11_	−4.1	359.1494, 341.1392, 137.0609	Matairesinoside (**1**)
4.46	(557.2019)	C_27_H_34_O_11_	−0.7	373.1662, 355.1558, 151.0770, 137.0614	Arctiin (**2**)
5.40	359.1494	C_20_H_22_O_6_	−0.3	341.1389, 137.0607	Matairesinol (**3**)
6.06	373.1653	C_21_H_24_O_6_	0.5	355.1551, 137.0611	Arctigenin (**4**)
6.51	715.2741	C_40_H_42_O_12_	−2.0	697.2642, 679.2540, 137.0606	Dimatairesinol (**5**)
6.99	729.2905	C_41_H_44_O_12_	−0.8	711.2792, 693.2686, 151.0761, 137.0606	Viridissimaol A (**6**)
6.99	729.2896	C_41_H_44_O_12_	−2.1	711.2792, 693.2689, 151.0760, 137.0608	Viridissimaol B (**7**)
7.40	741.2907	C_42_H_44_O_12_	−0.5	723.2800, 705.2671, 247.0977, 151.0762	Viridissimaol E (**8**)
7.49	743.3055	C_42_H_46_O_12_	−1.7	725.2947, 707.2844, 151.0764, 137.0605	Diarctigenin (**9**)
7.81	743.3060	C_42_H_46_O_12_	−1.1	725.2958, 707.2844, 151.0768, 137.0613	Conicaol A (**10**)

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
