# Peer review of "Comprehensive Characterization of Lignans from Forsythia viridissima by UHPLC-ESI-QTOF-MS, and Their NO Inhibitory Effects on RAW 264.7 Cells"

_molecules, 2019, doi:10.3390/molecules24142649_

Round 1
Reviewer 1 Report
L17 number agreement; replace “have” with “has”
L18 insert a connecting sentence about the study motivation between first and second sentence of the abstract. Is it known that lignans show anti-inflammatory activity?
L24 comment on the difference between lignin and lignin; compound 3 and 4 are lignans, aren’t they?
L22: introduce shortly why you investigate NO inhibitory effects
L25: introduce the abbreviations LPS and iNOS
L27-28: the last sentence is a week statement about the meaning of the study. Pls make the importance more clear.
Add a figure illustrating the chemical structures of the detected compounds
L60: delete “The”
L61: replace suspended with diluted
L65 number agreement “were”/was
L68: Rephrase e.g. “The lignans present in F. viridissima were identified by calculation of the molecular formula from exact mass and MSe spectra with the characteristic mass fragmentation patterns.”
L71 number agreement “its”/their
L73 replace fragmentation with fragment ions
L75 number agreement “molecular ions”
Always place m/z before the value of the m/z throughout the manuscript. Delete all “of” between m/z and their values
L77 pls explain the labeling of the chroms in the legend of fig s1. Moreover, the order of the sup figures seems awkward and should follow their appearance as reference in the manuscript.
All supp figures are labeled with “S” after the number, all references with “S” before the number, pls adjust
L78: Rephrase: ”The characteristic MS fragment of m/z 137 of matairesinol was also observed (Figure S7) in agreement with [16, 17].
L 84: do you refer to figure 1 here?
L86 rephrase: “The daughter ions of both the compounds also exhibited diagnostic ions at m/z 137 and 151.
Reduce size of the upper panel in Fig 1 to match the lower panel. Make the two panels visually more clear. Improve the legend: explain A and B, label the different panels etc. For better comprehension, highlight the two regions in the molecule where ring opening is suggested in the lower panel- according to your belief, does this happen in the gas phase or in solution? Moreover, you suggest a twofold charged species here?? And finally, was the suggested fragmentation confirmed or is this just a proposal of yours? I would assume, other cleavages would also be possible?
L95-96: Rephrase “In contrast, the observed fragmentation at m/z of 247 could occur due to the unsaturation of the position 7 and 8 followed by a rearrangement resulting in loss of the substituted benzene moiety (Figure 2) [16, 17].”
Fig 2: upper panel can be reduced in size and, anyway, is redundant with Fig 1. Lower panel first structure contains overlapping elements, pls improve. The proposed mechanism contains several H migration, what makes you think that the rearrangements are not simultaneously? Pls better illustrate the proposed mechanism using arrows to highlight the purpose of the figure
L102 number agreement “molecular ions”
L104 number agreement “spectra…were obtained”
Table 1: legend: “Compounds identified by molecular formulas calculated from the accurate mass and MSe fragments of UHPLC-ESI-QTOF-MS analyses.” Consider using “MSe” instead MS fragments for consistency
L109: number agreement “roots exert”
L111 explain selection of the cell line
L118 6,7 and 8 also show inhibition in a dose-dependent manner? Just not linear?
Fig3: shift graphs for inactive compounds to the supps. The legend is too short; e.g. explain the stars and color-coding of the bars and the numbers above the graph; axis labeling could be a bit more suggestive
Add a reasonable conclusion
L139 add a paragraph summarizing all used chemicals and suppliers
L144 “subfractions”
L149 you sure you used w/v not v/v; moreover, either of the two is written in italics
L154 0.0 L/h is “off”; missing e in “cone”; two modes- does that mean you´measured each sample twice?
L155 really 600.0 L/h? (check precision for given number of digits)
L157 delete “The”, missing e in encephaline, add lock mass in pos mode
L167 detail how the compounds were applied, did you change the medium? If not, how did you achieve homogenous distribution?
L172 comment on how valid the quantity of nitrite works as a proxy for the discussed NO; in fact, the whole manuscript never mentions that you in fact used a nitrite assay; how you ensure that we see the effect of NO not nitrite?
L176 provide reference for Griess reagent
References: check italic font for the Journal issues throughout the whole list. Check journal abbreviation ref 10
Author Response
We thank the Reviewers for the thoughtful comments. We tried to address those comments and we sincerely hope that the revisions will render this manuscript suitable for publication in Molecules.
Please see the attachment.

Reviewer 2 Report
The manuscript “Comprehensive characterization of lignans from Forsythia viridissima by UHPLC-ESI-QTOF-MS, and their NO inhibitory effects on RAW 264.7 cells” describes 10 lignans in Forsythia viridissima roots as well as their NO inhibitory effects on RAW 264.7 cells. Actually, all the ten compounds identified were recently reported (Huh et al., DOI: 10.1021/acs.jnatprod.8b00590) as constituents of Forsythia viridissima roots. In addition, there is a lack of information in the experimental section regarding the extraction and fractionation methodology used for the compounds identified or even how they were isolated for anti-inflammatory analysis. Finally, the anti-inflammatory activity of lignans, including of some of those identified in this manuscript is already described in the literature. Therefore, there is a considerably lack of novelty. I suggest reject it.
Author Response

(The authors gave the same response as above.)

Reviewer 3 Report
Jungmoo Huh et al. analyzed extracts of Forsythia viridissima to identify compounds with anti-inflammatory effects. They identified various lignans and determined their in vitro effects on NO inhibition and iNOS expression. The experiments have been carefully designed and performed.
Major compulsory revisions
The authors describe that fruits Forsythia viridissima of have been traditionally used for inflammatory conditions. Yet they investigated the roots of that plant. They should give a rational why they investigated the roots and discuss whether similar or different compounds have been identified in the fruits.
The authors performed elaborate and nice experiments. However, they do not provide sufficient detailed information to reproduce them. Please either give more experimental details or provide a reference where a detailed protocol can be found. Examples:
Line 60: the extract: please detail ratio of root and solvent, temperature, extraction time and conditions.
Line 121: immunoblotting: please detail whether wet or dry blotting was used, blotting conditions. A section “Immunoblotting” is not present in the “Materials and Methods” section. What was used as control? Which antibodies were used (provider/company?), which dilutions of the antibodies were used? Please revise.
Line 125: methylene chloride fraction: please detail extraction conditions. Was the solvent evaporated after extraction before it was used in cell culture? Was the residue reconstituted for the cell culture experiments? Which solvent was used for reconstitution?
Line 144 and following: which software was used for peak analysis?
Line 167: which solvent was used for the compounds? What was used as control in the cell viability assays?
Line 175: which solvent and concentration was used for LPS? What was the calibration range for the nitrite assay?
The discussion should put the results of the present study into the context of existing knowledge. So please explain whether the compounds you identified have been found in other plants / parts of plants before. Has an anti-inflammatory activity previously been described for these compounds? If so, in which assay?
Minor compulsory revisions
The expression in line 47 “Lignan is a representative secondary metabolite of F. viridissima.” is not precise. Lignan is not a single compound, lignans are a group of compounds. Please revise.
Line 52: the authors mention “numerous studies”. Please specify the type of studies: in vivo, in vitro, animal, human studies?
Figure 3: What does the first column in the nitrite figures represent? Do the columns represent means and standard deviations? Or standard error? Please provide the number of repetitions of the experiments (n= ?). The stars obviously indicate statistical significance, please explain (p< ?). Which statistical test was used? Please add the topic “Statistical analysis” in the “Materials and Methods” section.
Figure 4: Please provide the number of repetitions of the experiments (n= ?). Please detail in the “Materials and Methods” section how the densitometric analysis was done (software?).
Author Response

(The authors gave the same response as above.)

Round 2
Reviewer 3 Report
The authors responded appropriately to the comments and revised their manuscript accordingly. i find the manuscript suitable for publication now.